# Pre-Existing Immunity Predicts Response to First-Line Immunotherapy in Non-Small Cell Lung Cancer Patients

**DOI:** 10.3390/cancers16132393

**Published:** 2024-06-28

**Authors:** Anastasia Xagara, Maria Goulielmaki, Sotirios P. Fortis, Alexandros Kokkalis, Evangelia Chantzara, George Christodoulopoulos, Ioannis Samaras, Emmanouil Saloustros, Konstantinos Tsapakidis, Vasileios Papadopoulos, Ioannis S. Pateras, Vasilis Georgoulias, Constantin N. Baxevanis, Athanasios Kotsakis

**Affiliations:** 1Laboratory of Oncology, Faculty of Medicine, School of Health Sciences, University of Thessaly, 41110 Larissa, Greece; xagaraa@hotmail.com; 2Cancer Immunology and Immunotherapy Center, Saint Savas Cancer Hospital, 171 Alexandras Ave., 11522 Athens, Greece; mgoulielmaki@eie.gr (M.G.); sfortis1989@gmail.com (S.P.F.); costas.baxevanis@gmail.com (C.N.B.); 3Department of Medical Oncology, University General Hospital of Larissa, 41110 Larissa, Greece; alexkokkalis@hotmail.gr (A.K.); valiaxantzara@gmail.com (E.C.); gchristodoulopoulos@hotmail.gr (G.C.); jnsamaras@gmail.com (I.S.); esaloustros@yahoo.gr (E.S.); tsapakidisk@yahoo.com (K.T.); vasilispapadopoulos1@hotmail.com (V.P.); 4Second Department of Pathology, “Attikon” University Hospital, Medical School, National and Kapodistrian University of Athens, 15772 Athens, Greece; ipateras@med.uoa.gr; 5First Department of Medical Oncology, Metropolitan General Hospital, 15562 Athens, Greece; georgulv@otenet.gr

**Keywords:** NSCLC, pre-existing immunity, immunotherapy, PD-1, T cells, Tregs, MDSCs

## Abstract

**Simple Summary:**

The content as well as the status of immune cells before immunotherapy administration are indispensable for response. TAA-specific T cells have been associated with poor responses to chemotherapy. However, their role as predictive biomarkers for immunotherapy administration as well as their interplay with other immune system cells is not well understood. Our findings reveal that combined analysis of pre-existing immunity T cells of different immune parameters in treatment-naïve NSCLC patients leads to better prediction tools for immunotherapy response.

**Abstract:**

T-cell-mediated anti-tumoral responses may have significant clinical relevance as a biomarker for response to immunotherapy. The value of peripheral blood pre-existing tumor antigen-specific T cells (PreI^+^) as a predictive immunotherapy biomarker in NSCLC patients was investigated, along with the frequency of various circulating immune cells. Fifty-two treatment-naïve, stage III/IV NSCLC patients, treated with front-line immune checkpoint inhibitors (ICI)-containing regimens were enrolled. PreI was calculated as the percentages of CD3^+^IFNγ^+^ cells after in vitro co-cultures of PBMCs with peptides against four different Tumor-Associated Antigens (TAA). Immunophenotyping of peripheral blood immune cells was performed using multicolor flow cytometry. PreI^+^ T cells were detected in 44% of patients. Median overall survival (OS) was significantly higher in PreI^+^ patients compared to PreI^–^ patients (not reached vs. 321 days, respectively; *p* = 0.014). PreI^+^ patients had significantly higher numbers of possible exhausted CD3^+^CD8^+^PD-1^+^ cells and lower percentages of immunosuppressive Tregs compared to PreI^−^ patients. Additionally, patients with PreI^+^ and low numbers of peripheral blood M-MDSCs had a significant survival advantage compared to the rest of the patients. Thus, combining pre-existing tumor antigen-specific immunity before initiation of ICI in NSCLC patients with selected immune-suppressive cells could identify patients who have a favorable clinical outcome when treated with ICI-containing regimens.

## 1. Introduction

Lung cancer is the most lethal cancer type globally, and Non-Small Cell Lung Cancer (NSCLC) accounts for 85% of all cases [1]. The majority of patients are diagnosed with locally advanced or metastatic disease, thus leading to a poor prognosis [2]. Therapeutic options for stage III and IV patients include chemotherapy, radiotherapy, and targeted therapy. In addition, in recent years, immune checkpoint inhibitors (ICIs) have emerged as an important therapeutic option since they lead to significant improvement in overall survival (OS) and progression-free survival (PFS) [3]. Indeed, ICIs have revitalized cancer treatment since recent data indicate a 5-year survival rate of unselected stage IV NSCLC patients of about 20% and of 40% of patients with tumors expressing high PD-L1 levels [4]. Nonetheless, a significant number of patients fail to respond to ICIs or develop resistance and disease progression after an initial clinical response [5]. Therefore, there is an unmet need to accurately define the patients who may benefit from ICIs using novel predictive biomarkers.

The immune system plays a pivotal role in cancer elimination. T cells comprise the cornerstone of adaptive immunity and are indispensable in cancer progression and immunotherapy [6]. For efficient immunotherapy, CD8^+^ and CD4^+^ T cells should be able to recognize antigens presented by the major histocompatibility complex on cancer cells [7]. Along this line, immune cells should be able to retain their anti-tumor capacity by passing through tumor immunoediting to achieve durable responses [8]. Hence, deciphering the immune status prior to and during treatment may reveal novel and accurate predictive biomarkers for therapy response.

Pre-existing tumor-antigen-specific T cell detection may be superior in predicting response to ICI therapy. We have previously shown the safety and efficacy of a cryptic peptide vaccine against TERT polymerase [9,10,11,12], and we observed that NSCLC patients harboring TERT-specific CD8^+^ T cells before vaccination in their peripheral blood had a minimal response to chemotherapy, but there was a significant benefit in patients harboring low levels of CD8^+^ T cells in their tumors [10,12].

In the current study, the detection of peripheral blood PreI in patients with NSCLC was assessed after in vitro cultures of patients’ PBMCs with four known tumor-associated peptides, along with a detailed immunophenotypic analysis of peripheral blood immune effectors and suppressors. To our knowledge, this is the first study correlating pre-existing antigen-specific T cells with survival benefits in front-line immunotherapy for NSCLC patients.

## 2. Materials and Methods

### 2.1. Patients and Blood Collection

A total of 52 chemotherapy-naïve, histologically confirmed NSCLC patients were enrolled in this study. The patients’ median age was 70 years old (range: 48–86), 35 (67.3%) had stage IIIb disease and received definitive concurrent chemoradiation (cCRT) followed by durvalumab (PD-L1 inhibitor, Imfinzi, Astra Zeneca, London, UK), according to National Comprehensive Cancer Network (NCCN) guidelines; the remaining 17 patients (32.7%) with stage IV received combined chemotherapy and ICI as front-line treatment (Table 1, Appendix A). Additionally, 15 healthy donors (HD) were analyzed as matched controls. The following eligibility criteria were required for the stage III patients: age > 18 years, histologically confirmed diagnosis of NSCLC, unresectable clinical stage III, PS (ECOG) 0–2, PD-L1 expression in ≥1% of tumor cells (according to the durvalumab indication), and disease control (CR, PR, SD) after completion of CRT, based on the RECIST 1.1. criteria.

Peripheral blood [20 mL in K_2_ ethylenediaminetetraacetic acid (EDTA; BD Biosciences, Heidelberg, Germany)] was obtained before the administration of the first ICI cycle (baseline sample). This study has been approved by local ethics and scientific committees (32710/3-8-20) and consequently was performed according to the Ethical Principles for Medical Research Involving Human Subjects of the World Medical Association Declaration of Helsinki. All patients and HD provided written informed consent to participate in the study.

### 2.2. Lymphocyte Isolation

For the isolation of peripheral blood mononuclear cells (PBMCs), Hypaque-1077 (Sigma-Aldrich, St. Louis, MO, USA) was used. Cells were then re-suspended in RPMI-1640 medium (Biosera, Nuaille, France) that was supplemented with fetal bovine serum (10%, heat-inactivated) (Gibco, Grand Island, NY, USA) and penicillin and streptomycin solution (1%) (Solarbio, Beijing, China).

### 2.3. Pre-Existing Immunity Detection and Analysis

PBMCs from the patients and HD were co-cultured with peptide mixtures from the tumor associated antigens TERT, MAGEA1, Survivin, and NY-ESO-1 (JPT Peptide Technologies, Berlin, Germany) according to our previous experiments (Vetsika 2012). More specifically, 0.5 × 10^6^ PBMCs were cultured with 10 μM of each peptide mixture or 100 ng/mL Staphylococcal enterotoxin B from Staphylococcus aureus (SEB) (Sigma-Aldrich/Merck, Darmstadt, Germany) as a positive control or DMSO (Sigma-Aldrich/Merck) as a negative control for 24 h. 20 h prior to harvesting, the monensin-containing Golgi Stop (BD Bioscience Pharmingen, San Diego, CA, USA) solution was added according to the manufacturer’s instructions. All co-cultures were performed in an incubator with 5% CO_2_ at 37 °C in triplicates using RPMI medium supplemented with 5% HS, 10% FBS, and 1% p/s for 24 h, and cells were harvested, washed with 1X PBS (Capricorn scientific, Ebsdorfergrund, Germany), and stained with antibodies against CD3, CD4, and CD8, as we described before [13]. Intracellular staining for anti-IFNγ FITC was followed using a fixation/permeabilization kit (BD Cytofix/Cytoperm, BD Biosciences, Franklin Lakes, NJ, USA) according to the manufacturer’s instructions (Appendix A).

### 2.4. Flow Cytometry Analysis

Detection of surface markers on PBMCs was performed using the following anti-human fluorochrome-conjugated monoclonal antibodies for T cells: anti-CD3 PE-Cy7; anti-CD8 APC-Cy7; anti-CD4 BV510; anti-CD45RO APC; anti-CD45RA PE; anti-PD-1 PerCPCy5.5; anti-CCR7 FITC; (gating strategy for T cells in see Appendix A) for Tregs: anti-CD3 PE-Cy7; anti-CD4BV510; anti-FOXP3FITC; anti-CTLA-4APC;anti-CD25 PerCPCy5.5;anti-CD127BV421; (gating strategy for Tregs in Appendix A) for MDSCs: anti-CD33 APC-Cy7; anti-CD11bPE;anti-HLA-DRPE-Cy7;anti-Lin PerCPCy5.5 (anti-CD3 PerCPCy5.5; anti-CD19PerCPCy5.5; anti-CD56PerCPCy5.5; anti-CD4PerCPCy5.5; anti-CD16PerCPCy5.5); anti-CD14BV510; anti-CD15APC; anti-iNOSFITC (gating strategy for MDSCs in Appendix A) (all antibodies were purchased from Biolegend, San Diego, CA, USA). Staining was performed using FACS buffer that contained 1% BSA, and the incubation was 30 min on ice in the dark, as we described previously [14]. Acquisition and analysis were performed by BD FACSChorus Software on a Melody flow cytometer (BD Biosciences, Heidelberg, Germany). For analysis of the T cell subsets, gates were assigned to the lymphocytic population. For each measurement, 10^6^ single events were analyzed. For the negative control, unstained cells were used. To set up the gates, FMO-stained cells were used.

### 2.5. Statisical Analysis

For statistical analysis, GraphPad Prism version 10 (GraphPad Institute Inc., San Diego, CA, USA) was used. For overall survival (OS), the time from enrollment to the study until death from any cause or the last time follow-up that the patient was reported alive was used. For progression-free survival (PFS), the time between enrollment and disease relapse or death, whatever occurred first, was used. To correlate immune cell phenotypes with patients’ clinical outcomes, Kaplan–Meier analysis was used. The groups were compared using the log-rank test. Nonparametric Mann–Whitney *U* test was used to determine differences between groups. Cutoffs used to divide immune cell (IC) percentages into low and high were defined from receiver operating characteristic (ROC) curves. Different clinical parameters with survival were correlated using Cox-regression analysis. For significant differences and associations, a *p* < 0.05 was used. All *p* values were two-sided.

## 3. Results

### 3.1. Pre-Existing TAA-Specific T Cells in the Circulation of NSCLC Patients

To access the levels of circulating TAA-reactive T cells in NSCLC patients, we performed co-cultures of PBMCs with peptide mixes against four different, highly relevant to NSCLC, TAAs called TERT, MAGEA1, Survivin, and NY-ESO-1. For positive controls, patients’ PBMCs were cultured with SEB, while for negative controls, DMSO was used. To set up the cutoffs, co-cultures with matched healthy donor PBMCs were performed (Figure 1A and Appendix A). Based on the detection of IFNγ-expressing T cells against the peptides from any TAA, 44% (23/52) of the patients were considered to have PreI^+^, whereas, no IFNγ-expressing T cells against the same antigens could be revealed in the remaining 29 patients (56%; PreI^−^) (Figure 1A,B).

The percentages of PreI^+^ patients, referred to as the number of different peptides recognized by their T cells, were distributed as follows: 5.8% (3/52) recognized one TAA, 1.9% (1/52) two TAAs, 7.7% (4/52) three TAAs, and most of them 29% (15/52) recognized all four TAAs (Figure 1B). Phenotypic characterization of PreI^+^ T cells indicated that in 65% (15/23) of the patients, the IFNγ-secreting cells were CD3^+^CD4^+^ and CD3^+^CD8^+^; conversely, in 13% (3/23) and 22% (5/23) of the patients, only CD3^+^CD4^+^ and CD3^+^CD8^+^ were shown to secrete IFNγ, respectively (Figure 1C).

### 3.2. Pre-Existing TAA-Specific T Cells and Clinical Response

The clinical relevance of pre-existing TAA-specific T cells in the peripheral blood was assessed in both PreI^+^ and PreI^−^ patients. Figure 1D indicates that the median OS was significantly higher in PreI^+^ stage IIIb patients who received ICI compared to PreI^−^ (not reached vs. 390 days, respectively, HR 2.74; *p* = 0.028); nevertheless, there was no difference regarding the median PFS (294 days vs. 323 days; *p* = 0.94). Additionally, there was a statistically significant longer median OS in PreI^+^ all-stage patients irrespective of the used immunotherapy regimen compared to PreI^−^ patients (not reached vs. 321 days, HR 2.93; *p* = 0.014); however, once again, there was no difference in terms of PFS between the two groups of patients (*p* = 0.38). Cox regression analysis revealed the absence of any correlation between the different clinical parameters and patients’ survival (Appendix A). These results indicate that pre-existing TAA-specific T cells in the circulation of NSCLC patients receiving ICI therapy as front-line treatment could serve as a prognostic biomarker for survival.

### 3.3. Immune Effectors in the Circulation of Pre-Existing Immunity Patients

To further understand the relevance of PreI^+^ TAA-specific T cells, a detailed immunophenotypic analysis of peripheral blood immune cells was performed. Analysis of the different T cell subpopulations [naïve (CD45RA^+^CD45RO^−^CCR7^+^), effectors (CD45RA^+^CD45RO^−^CCR7^−^), central memory (CD45RA^−^CD45RO^+^CCR7^+^), and effector memory (CD45RA^−^CD45RO^+^CCR7^−^)] that were positive or negative for PD-1 in both CD3^+^CD4^+^ and CD3^+^CD8^+^ cells could not reveal any significant difference between patients’ groups (Appendix A). Importantly, PreI^+^ patients had significantly higher percentages of PD-1-expressing CD3^+^CD8^+^ (*p* = 0.0078) but not CD3^+^CD4^+^ (*p* = 0.555) T cells compared to PreI^−^ patients (Figure 2A,B).

To correlate CD4^+^ and CD8^+^ T cells in circulation with survival, patients were dichotomized into low and high using the cutoffs defined by ROC curves (Table 2). PFS and OS analysis indicated a significantly higher survival probability in patients harboring high levels of CD3^+^CD4^+^ T cells that express PD-1 in circulation (log rank 0.0089) (Table 2).

Additionally, survival analysis revealed that PreI^+^ patients harboring high levels of CD3^+^CD8^+^PD-1^+^ or CD3^+^CD4^+^PD-1^+^ had a significantly longer survival compared to PreI^−^patients (Figure 2C,D).

### 3.4. Peripheral Blood Immune Suppressor Cells in PreI^+^ Patients

The percentages of immunosuppressive Tregs and MDSCs were also examined. Basic Tregs (CD3^+^CD4^+^CD25^+^CD127^−^FOXP3^+^) were not significantly different between PreI^+^ and PreI^−^ (*p* = 0.118) patients (Figure 3A; Table 3); however, the percentages of CTLA-4^+^ Tregs (CD3^+^CD4^+^CD25^+^CD127^−^FOXP3^+^CTLA-4^+^) were found to be marginally lower in PreI^+^ patients (*p* = 0.049) compared to PreI^−^ patients (Figure 3A; Table 3). Similarly, the frequency of monocytic MDSCs, expressing or not iNOS, was not different among the two patients’ groups (Figure 4A; Table 3).

The correlation of different immunosuppressive phenotypes, grouped into high and low, with survival outcomes did not reveal any significant difference (Table 4). As PreI^+^ patients were found to have lower percentages of CTLA-4 Tregs in circulation, we next grouped the patients as follows: Group 1: High CTLA-4 Tregs with PreI^+^ (*n* = 13), Group 2: High CTLA-4 Tregs with PreI^−^ (*n* = 21), Group 3: Low CTLA-4 Tregs with PreI^+^ (*n* = 9), and Group 4: Low CTLA-4 Tregs with PreI^−^ (*n* = 9). Kaplan-Mayer curves indicated a significant overall survival benefit in patients in group 3 compared to patients in group 4 (*p* = 0.042, med: undefined vs. 284 days, HR 0.14) but not PFS (*p* = 0.562) (Figure 3B).

Additionally, the patients were also grouped according to high or low percentages of CD14^+^CD15^+^ M-MDSCs and their pre-existing immunity T cell status as follows: Group A: High M-MDSCs with PreI^+^ (*n* = 10), Group B: High M-MDSCs with PreI^−^ (*n* = 9), Group C: Low M-MDSCs with PreI^+^ (*n* = 12), and Group D: Low M-MDSCs with PreI^−^ (*n* = 19). Survival analysis revealed a significant benefit in patients with PreI^+^ harboring low levels of M-MDSCs (Group C) compared to the other three groups (Group C vs. Group A: *p* = 0.050, med: undefined vs. 411 days, HR0.16; Group C vs. Group B: *p* = 0.009, med: undefined vs. 321 days, HR 0.10; Group C vs. Group D: *p* = 0.039, med: undefined vs. 329 days, HR 0.25) (Figure 4B).

## 4. Discussion

In the present study, we evaluated the predictive value of peripheral blood pre-existing TAA-specific T cells in locally advanced and metastatic NSCLC patients receiving front-line ICI-containing therapeutic regimens. We showed that the detection of pre-existing TAA-specific T cells in the peripheral blood before the initiation of systemic immunotherapy may have a significant prognostic value in the ICI response. Indeed, patients with PreI indicated significantly longer OS compared to patients without PreI, but not PFS. It is known that PFS has a suboptimal predictive value for diseases with longer survival post-progression, such as NSCLC, especially for immunotherapy treatments that may alter tumor growth kinetics rather than solely act via direct cytotoxicity [15,16]. Moreover, we analyzed and compared the major immune populations in circulation, such as effector subpopulations of CD3^+^CD8^+^ and CD3^+^CD4^+^ T cells and immunosuppressive Tregs and MDSCs, between patients with PreI^+^ and PreI^−^. We observed that PreI^+^ patients that respond to ICI-containing regimens bear significant percentages of CD3CD8PD-1+ T cells associated with an exhausting phenotype compared to PreI^−^. Moreover, patients with high percentages of PD-1^+^ T cells and PreI^+^, as well as patients with PreI^+^ and low percentages of either Tregs or MDSCs, experienced significantly longer survival compared to PreI^−^ patients. Grouping of patients with PreI^+^ and low percentages of either Tregs or MDSCs indicated survival advantage. Taken together, these observations strongly suggest the presence of a favorable peripheral blood immune status in NSCLC patients with pre-existing immunity.

The impact of functional circulating TAA reactive T cells on survival has already been mentioned previously [17], as well as by our group after vaccination with a cryptic TERT vaccine [9,10,11,12,18,19,20] in different cancer types, including NSCLC. It is worth mentioning that these TAA were chosen as they show strong in vivo immunogenicity, they expressed at high levels compared with others in NSCLC, and additionally, their expression is not mutually exclusive [21,22]. The functionality of these cells may indicate a general immune reactivity against TAA in these patients that could be attributed to the relatively high neoantigen loads and higher mutational status. Indeed, we observed that the majority of the PreI^+^ patients (15/23) harbored T cells reactive to all four TAA peptides. Other studies have also shown a high probability of pre-existing T cell reactivity against multiple TAAs in their peripheral blood [17,23]. Unfortunately, in the current study, the patients’ tumor mutational load could not be assessed due to tissue unavailability.

As we speculated that PreI^+^ patients may be more immunogenic than PreI^−^ patients, we performed a comparative analysis of different subtypes of peripheral blood, including naïve and differentiated T cells. Univariable analysis revealed significantly higher percentages of CD3^+^CD8^+^PD-1^+^ possible exhausted T cells in PreI^+^ patients compared to PreI^−^; conversely, there was no significant difference regarding the frequency of other T cell subsets such as naïve, effector, effector memory, or central memory T cells that express or do not express PD-1.

It has been shown in melanoma patients that CD8^+^PD-1^+^ but not CD8^+^PD-1^−^ T cells in the peripheral blood had increased reactivity to TAA such as NY-ESO1 and MAGE in vitro, indicating enrichment of tumors with tumor-specific immune cells that might be exhausted [24]. In our study, we observed a significantly higher percentage of peripheral blood CD8+PD-1^+^T cells that was related to longer survival upon immunotherapy, a finding that may indicate reinvigoration of these cells. However, due to the low numbers of pre-existing immune T cells, we were not able to characterize the expression of PD-1 in this specific population.

Detection of functional pre-existing immunity T cells also indicate that these cells can resist the immunosuppressive activity of Tregs and MDSCs [25]. Tregs have the ability to repress natural anticancer immune responses and are considered important players for poor responses to ICI [26]. We and others have shown that high numbers of Tregs are associated with a worse prognosis in NSCLC patients [27,28]. In the current study, high levels of patients’ Tregs could not predict response to ICI; however, there was a trend towards poorer survival for CD3^+^CD4^+^FOXP3^+^ Tregs (*p* = 0.063); a possible explanation could be the low number of patients analyzed in the current study. In addition, patients with PreI^+^ had significantly lower levels of CTLA-4 Tregs compared to PreI^−^ patients. Importantly, patients with PreI^+^ and low levels of immunosuppressive CTLA-4 Tregs had a survival benefit compared to PreI^−^ patients. This finding is reported for the first time and supports our hypothesis for a favorable immune status in patients with pre-existing immunity.

MDSCs are a heterogenous group of immature myeloid cells that exert multiple immunosuppressive functions, including inhibition of T cell activation and Tregs expansion [29,30]. In humans, there are two distinct subpopulations named granulocytic MDSCs (G-MDSCs) and monocytic MDSCs (M-MDSCs) [31]. In the current study, G-MDSCs were not studied due to technical reasons, as mentioned above. In NSCLC, M-MDSCs are considered a poor prognostic factor of response [32] and have also been correlated with shorter PFS and OS in patients receiving frontline chemotherapy [33,34]. Their predictive value in patients receiving first-line immunotherapy has been described by other groups [35]. In the current study, high levels of (CD14^+^CD15^+^) M-MDSCs were also associated with poorer survival upon ICI treatment, but there were no significant differences in their percentages between PreI^+^ and PreI^−^ patients. Importantly, when we grouped the patients into those having PreI and having low levels of M-MDSCs (CD14^+^CD15^+^), we observed a significant survival benefit compared to PreI^−^ patients. It should be mentioned that all but one patient were alive at the cutoff date of the follow-up. In line with our findings, other groups have shown that older breast cancer patients (range 65–87 years) bearing high levels of Her2+ pre-existing T cells have low percentages of MDSCs in circulation and, more importantly, have a significant survival advantage [36]. These results are reported for the first time in NSCLC and highlight the complexity of immune interactions, indicating the importance of studying different immune components simultaneously.

## 5. Conclusions

In conclusion, in the current study, a comprehensive analysis of different peripheral blood immune parameters of locally advanced/metastatic NSCLC patients was performed. We revealed that patients with pre-existing TAA-specific T cells have a survival benefit when treated with frontline ICI-containing regimens. By analyzing different immune populations, we demonstrate that these patients have high percentages of possible exhausted PD-1^+^ T cells in their peripheral blood. Importantly, we could detect a sub-category of PreI^+^ patients with significant survival advantage when treated with ICI, those that bear low levels of Tregs and MDSCs immune suppressive cells. MDSCs seem to play a fundamental role since all but one patient were alive at the end of follow up. Thus, combined analysis of different immune cell subpopulations may offer stronger predictive biomarkers that may help in the identification of those patients that will benefit more from ICI therapy.

## Figures and Tables

**Figure 1 cancers-16-02393-f001:**
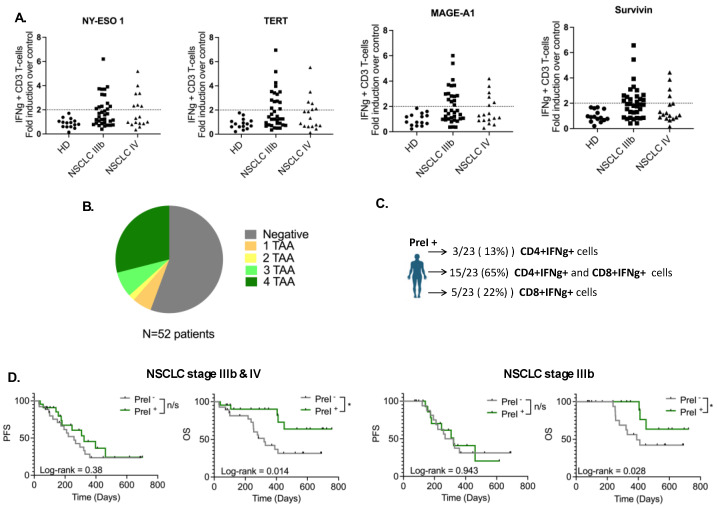
Pre-existing TAA-specific T cells in NSCLC patients. (**A**) Graphs depicting fold induction over negative control of IFNg+CD3+ T cells in healthy donors and NSCLC patients (**B**) pie chart showing the number of different TAAs recognized by patients T cells (**C**) percentages of patients with CD3^+^CD4^+^ and CD3^+^CD8^+^ TAA-specific T cells in circulation (**D**) Kaplan–Meier plots of PFS and OS in patients separated as PreI^−^ and PreI^+^. HD: healthy donors (*n* = 15); NSCLC IIIb: patients with IIIb disease (*n* = 35); NSCLC IV: patients with IV disease (*n* = 17); n/s: non-significant; * *p* < 0.05.

**Figure 2 cancers-16-02393-f002:**
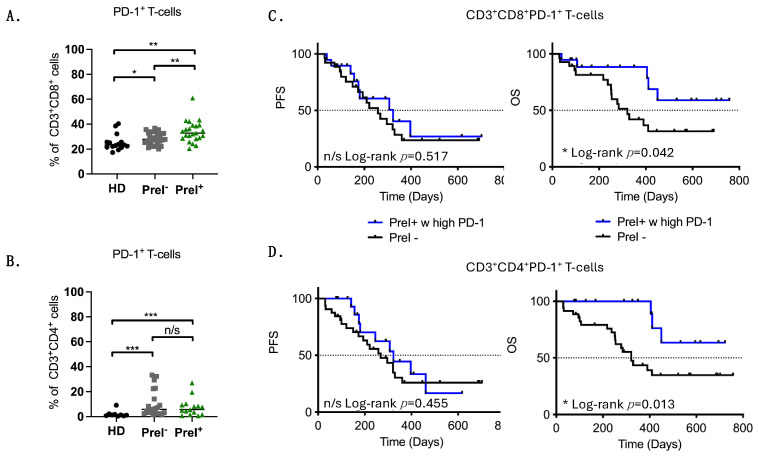
CD8^+^ and CD4^+^ T cells in the circulation of PreI^+^ NSCLC patients. Graphs show the percentages of PD-1^+^ in (**A**) CD8 and in (**B**) CD4 T cells; (**C**) Kaplan–Meier plots of PFS and OS in patients separated in PreI^+^ with high percentages of CD3^+^CD8^+^PD-1^+^ T cells; and (**D**) CD3^+^CD4^+^PD-1^+^ T cells vs. PreI^−^ patients. HD: healthy donors (*n* = 15); PreI^−^: patients without TAA pre-existing T cells (*n* = 29); PreI^+^: patients with TAA pre-existing T cells (*n* = 23); n/s: non-significant; * *p* < 0.05; ** *p* < 0.001; *** *p* < 0.0001. Unpaired Student *t* test.

**Figure 3 cancers-16-02393-f003:**
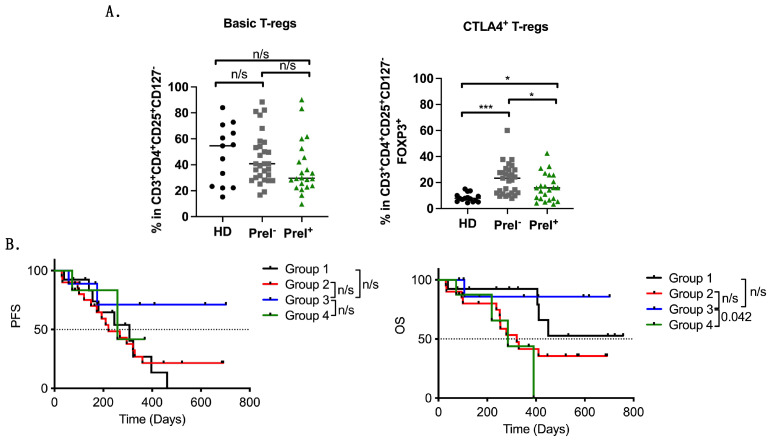
Immunosuppressive Treg cell populations in NSCLC patients. Graphs depicting the percentages of (**A**) basic and CTLA-4^+^ Tregs (**B**) Kaplan–Meier plots of PFS and OS in patients separated as follows: Group 1: High CTLA-4 Tregs with PreI^+^ (*n* = 13), Group 2: High CTLA-4 Tregs with PreI^−^ (*n* = 21), Group 3: Low CTLA-4 Tregs with PreI^+^ (*n* = 9), and Group 4: Low CTLA-4 Tregs with PreI^−^ (*n* = 9). n/s: non-significant; * *p* < 0.05; *** *p* < 0.0001. Unpaired Student *t* test.

**Figure 4 cancers-16-02393-f004:**
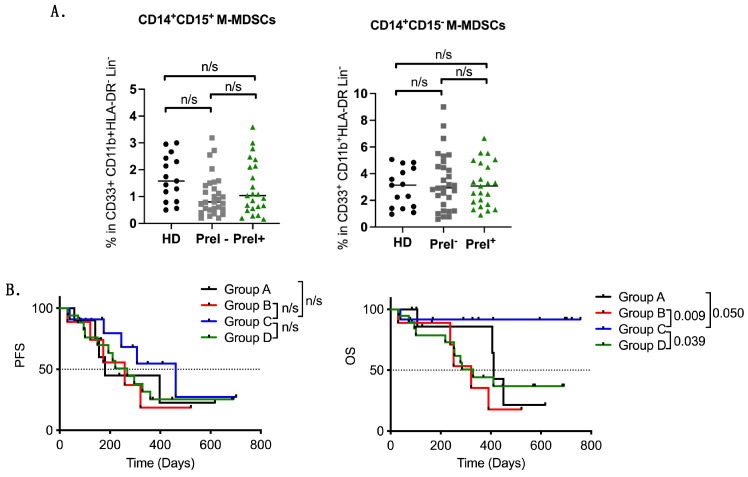
Immunosuppressive MDSC cell populations in NSCLC patients. Graphs depicting the percentages of (**A**) M-MDSC subpopulations and (**B**) Kaplan–Meier plots of PFS and OS in patients separated as follows: Group A: High M-MDSCs with PreI^+^ (*n* = 10), Group B: High M-MDSCs with PreI^−^ (*n* = 9), Group C: Low M-MDSCs with PreI^+^ (*n* = 12), and Group D: Low M-MDSCs with PreI^−^ (*n* = 19). n/s: non-significant. Unpaired Student *t* test.

**Table 1 cancers-16-02393-t001:** NSCLC patient’s characteristics.

		Stage IIIb (*n* = 35)	Stage III & IV (*n* = 17)	All Patients (*n* = 52)
Characteristics	Sub-Categories	Values	Values	Values
Median age		70 years (range 48–86 years)	70 years (range 52–82 years)	70 years (range 48–86 years)
Gender	Male	28 (80%)	12 (70.5%)	40 (77%)
	Female	7 (20%)	5 (29.5%)	12 (23%)
Stage	IIIb	35 (100%)	0 (0%)	35 (67%)
	III (other than IIIb)	0 (0%)	10 (59%)	10 (19%)
	IV	0 (0%)	6 (35%)	6 (12%)
	Unknown	0 (0%)	1 (6%)	1 (2%)
Location of primary tumor	Left lung	12 (34%)	4 (23.5%)	16 (31%)
	Right lung	22 (63%)	10 (59%)	32 (61.5%)
	Both lungs	1 (3%)	0 (0%)	1 (2%)
	Unknown	0 (0%)	3 (17.5%)	3 (5.5%)
Histological Type	Adenocarcinoma	17 (49%)	9 (53%)	26 (50%)
	Squamous	18 (51%)	6 (35%)	24 (46%)
	Unknown	0 (0%)	2 (12%)	2 (4%)
Smoking Status	Never	4 (11.5%)	0 (0%)	4 (8%)
	Former	20 (57%)	3 (17.5%)	23 (44%)
	Curent	11 (31.5%)	10 (59%)	21 (40%)
	Unknown	0 (0%)	4 (23.5%)	4 (8%)
	<40 pack year	4 (11%)	5 (30%)	9 (17%)
	40–80 pack year	13 (37%)	2 (11.5%)	15 (29%)
	>80 pack year	8 (23%)	2 (11.5%)	10 (19%)
	Unknown	10 (29%)	8 (47%)	18 (35%)

**Table 2 cancers-16-02393-t002:** Associations between different T cell phenotypes and clinical outcomes in treatment-naïve NSCLC patients. Bold *p* value indicates statistically significant difference.

					PFS			OS		
	T-Cell Populations	ROC Cut Off		*n*	Median	95% HR CI	*p* Value	Median	95% HR CI	*p* Value
% in CD3+CD8+	PD1	25	High	40	268	0.437 to 2.173	0.949	411	0.445 to 3.100	0.744
			Low	12	296			390		
% in CD3+CD8+ CD45RA+CD45RO-	Tnaive (RA+RO-CCR7^+^)	55	High	17	397	0.250 to 1.477	0.271	und	0.303 to 2.575	0.821
			Low	35	258			411		
	Teff (RA+RO-CCR7^−^)	40	High	36	268	0.273 to 2.295	0.668	450	0.150 to 1.718	0.276
			Low	16	329			284		
% in CD3+CD8+ CD45RA-CD45RO+	Tcm (RA-RO+CCR7^+^)	70	High	16	245	0.390 to 4.029	0.704	und	0.283 to 3.310	0.958
			Low	36	296			411		
	Tem (RA-RO+CCR7^−^)	26	High	35	307	0.141 to 1.380	0.159	450	0.126 to 1.152	0.087
			Low	17	258			284		
% in CD3+CD4+	PD1	2.2	High	35	321	0.202 to 1.187	0.114	und	0.093 to 0.712	0.0089
			Low	17	221			278		
% in CD3+CD4+ CD45RA+CD45RO-	Tnaive (RA+RO-CCR7^+^)	87	High	17	268	0.437 to 1.373	0.273	329	0.314 to 1.147	0.271
			Low	35	296			450		
	Teff (RA+RO-CCR7^−^)	13	High	37	321	0.249 to 1.332	0.197	und	0.237 to 1.552	0.297
			Low	15	268			329		
% in CD3+CD4+ CD45RA-CD45RO+	Tcm (RA-RO+CCR7^+^)	55	High	27	221	0.485 to 2.214	0.926	411	0.328 to 1.943	0.620
			Low	25	307			390		
	Tem (RA-RO+CCR7^−^)	40	High	26	296	0.463 to 2.127	0.984	450	0.509 to 3.00	0.637
			Low	26	307			411		

**Table 3 cancers-16-02393-t003:** Statistical comparison of Tregs and MDSC populations between patients with and without PreI. Bold *p* value indicates statistically significant difference.

	T Reg Cells		
CD3CD4FOXP3		PreI^−^	PreI^+^
	Mean	8.39	7.76
	Std.Error	1.106	1.163
	*p*-value	0.678	
CD25+CD127−	Mean	5.55	5.91
	Std.Error	0.808	0.790
	*p*-value	0.397	
Basic Tregs CD25+CD127-FOXP3+	Mean	44.97	37.57
	Std.Error	3.622	4.392
	*p*-value	0.118	
CTLA4+ Tregs CD25+CD127-FOXP3+CTLA4+	Mean	22.55	16.29
	Std.Error	2.19	2.16
	*p*-value	**0.049**	
	MDSCs		
CD14+CD15^−^ M-MDSCs (%in CD33+CD11b+HLA-DR-Lin-)		PreI^−^	PreI^+^
	Mean	3.38	3.18
	Std.Error	0.34	0.39
	*p*-value	0.713	
CD14+CD15- iNOS+ M-MDSCs (%in CD33+CD11b+HLA-DR-Lin- CD14+CD15-)	Mean	24.60	22.29
	Std.Error	3.085	3.038
	*p*-value	0.551	
CD14+CD15+ M-MDSCs (%in CD33+CD11b+HLA-DR-Lin-)	Mean	1.069	1.327
	Std.Error	0.141	0.207
	*p*-value	0.294	
CD14+CD15+ iNOS+ M-MDSCs (%in CD33+CD11b+HLA-DR-Lin- CD14+CD15+)	Mean	35.08	34.54
	Std.Error	2.59	3.59
	*p*-value	0.653	

**Table 4 cancers-16-02393-t004:** Associations of different Treg and MDSC phenotypes and clinical outcomes of treatment-naïve NSCLC patients. Bold *p* value indicates statistically significant difference.

Cell Populations	ROC Cut Off		*n*	Median	95% HR CI	*p* Value	Median	95% HR CI	*p* Value
CD3CD4FOXP3	5	High	31	321	0.254 to 1.198	0.133	450	0.173 to 1.066	0.068
		Low	21	210			390		
CD25+CD127-	2.7	High	38	321	0.384 to 1.977	0.742	und	0.381 to 2.288	0.882
		Low	14	201			411		
Basic Tregs	43	High	18	268	0.572 to 2.625	0.600	411	0.392 to 2.209	0.871
		Low	34	296			und		
CTLA4+ Tregs	11	High	34	268	0.908 to 4.680	0.083	411	0.476 to 3.229	0.658
		Low	18	und			und		
CD14+CD15- M-MDSCs	2.3	High	35	268	0.444 to 2.18	0.967	411	0.387 to 2.404	0.938
		Low	17	296			und		
CD14+CD15- iNOS+ M-MDSCs	19	High	24	268	0.707 to 3.163	0.291	405	0.665 to 3.713	0.302
		Low	28	321			und		
CD14+CD15+ M-MDSCs	1.2	High	18	173	1.064 to 6.983	**0.0047**	321	1.118 to 7.617	**0.0094**
		Low	34	329			und		
CD14+CD15+ iNOS+ M-MDSCs	39	High	21	307	0.386 to 1.745	0.610	und	0.339 to 1.902	0.619
		Low	31	258			405		

## Data Availability

The data presented in this study are available upon request from the corresponding authors.

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
