# Peer review of "Pre-Existing Immunity Predicts Response to First-Line Immunotherapy in Non-Small Cell Lung Cancer Patients"

_cancers, 2024, doi:10.3390/cancers16132393_

Round 1
Reviewer 1 Report
Comments and Suggestions for Authors
The authors described a very interesting and important topic in this paper. Immunotherapy for lung cancer patients is a very important issue these days, and there are many areas that need to be studied. This translational research covered various areas of immunotherapy for lung cancer patients.
1. The biggest weakness of this study is the small sample size. When the sample size is small, it is desirable to analyze only the results that can be confirmed with the small sample size. However, because this study attempted to analyze too many results with a small sample size, the overall quality of the paper deteriorated. If authors write a paper by analyzing only the content that can be confirmed with a sample size of 52 people, the quality of the paper will be further improved.
2. The prognosis of patients who received CCRT and patients who received only chemotherapy is very different. When analyzing PFS and OS, it is better to analyze the two patient groups separately. Table 2, Figure 2, lines 196-204 correspond to the content pointed out above.
3. The size of the Figures is too small. The letters in the Figures are also difficult to see.
4. Line 212-250.
Even though the sample size is small, the authors performed the analysis after segmenting the patient group too much, so the reliability of the statistical analysis is very low.
Key comments:
If authors want to keep the format and analysis method of this paper, increase the sample size and analyze again.
If authors want to maintain the current sample size, perform the analysis without segmenting the patient group too much.
Reviewer 2 Report
Comments and Suggestions for Authors
Lung cancer is one of the major causes of death and is hard to cure. In the manuscript „ Pre-existing Immunity predicts response to first-line Immuno-therapy in Non-Small Cell Lung Cancer patients” Xagara et al report that lung cancer patients undergoing ICI-based therapy will have a better overall survival (OS) if they show tumor-associated antigen (TAA)-specific T cells in their blood.
The topic of the study is interesting. The manuscript is well organized and written.
I raise the following points:
1. The authors should discuss why TAA-reactive blood T cells can predict a better overall survival (OS), but cannot predict progression-free survival (PFS)!
2. Please give the percentages of IFN-gamma secreting (CD4+/CD8+) T cells for all TAA used (how much is “2fold induction over control”?)
3. Ki67 is mentioned in Table2, but is not mentioned in Material&Methods or in Results.
4. Please explain in Table 2, which stained cells correspond to naïve, central memory, effector memory and terminal differentiated effector memory cells! Furthermore, the gating strategy should be given in the manuscript.
5. Please give the gating strategy for the determination of myeloid suppressor cells. I do not understand why healthy control persons have the same amount of suppressive monocytes as the patients (30% HLA-DR-negative?). This observation is contradictory to former studies, please compare your numbers of suppressive monocytes with results of other groups in the discussion!
6. The y-axis of Fig.3B gives “HLA-negative” monocytes and means HLA-DR-negative?
7. Which antibodies do you use for Lin-negative (page 4, line 123)?
8. In Fig.3 “Basic Tregs” are 40-50% of CD4+ T cells, also for healthy control persons. Normal Treg levels are 7±7 % of CD4+ T cells in peripheral blood. Why the Treg levels of this study are so high? Please show the gating strategy and discuss your results in context to other papers!
9. Please discuss, why a high proportion of naïve T cells is without a survival effect in your study (Table 2), but corresponds to better patients’ survival in other reports!
10. In Fig.2 the authors compare high-PD1+ Pre I+(blue line) with all Pre I- patients (black line), why they do not compare PD1+ Pre I+ with PD1+ PreI- patients?
11. Could the authors give all patient numbers in the high and low groups in Table 2 as well as in Table 4!
12. Please give the unit of measurement in Table 3. What “percentages” are meant?
13. Please discuss the usefulness of TERT, MAGEA1, Survivin and NY-ESO-1 for lung cancer! Are there other TAAs which have to be considered in diagnostics?
Round 2
Reviewer 1 Report
Comments and Suggestions for Authors
It is much better than the initial paper.
Although the revised paper still has weaknesses due to the small sample size, the reliability of the results has increased significantly by changing the statistical analysis method.
Author Response
We would like to thank the reviewer for the comments.
Reviewer 2 Report
Comments and Suggestions for Authors
The revised manuscript „ Pre-existing Immunity predicts response to first-line Immuno-therapy in Non-Small Cell Lung Cancer patients” of Xagara et al. has undoubtedly improved. However, the following points remain:
1. The MDSC numbers in Fig.4 have been changed (from 20-30% CD14+CD15- in the former manuscript to 3% in the new version), but the MDSC numbers in Table 3 have remained the same (30% CD14+CD15- M-MDSCs). Also in Table 4, the old cutoff between low and high of 25% is given. What is the explanation? Please make sure your results are consistent!
2. Please explain the MDSC gating in more detail or cite the reference for the MDSC gating! Are you sure that you have MDSC with gating the CD15+CD14+ cells? The gating of human MDSC in flow cytometry has been reviewed recently (PMID: 38555150). Could this help you with the gating of MDSC?
3. Please give in Table 3 all units of measurement in detail (CD14+CD15- M-MDSC as % of PBMC?, INOS+MDSC as % of what…)
4. Please mention in the text all of the supplementary figures for the gating of cells!
5. The new Suppl. Figures have to be mentioned in the "Suppl. Materials", line 345-349
6. In the discussion, the references 100, 101 (line 261), 102, 103 (line 278) are mentioned. However, these numbers cannot be found in the chapter “References”, where 32 references have been shown.
7. Please give the results for the naïve, effector, central memory and effector memory cells in patients and control group as “% of CD8”or “% of CD4” in a separate Figure/Table.
8. Please use also in Table 2 the unit “% of CD8” and “% of CD4”. The unit of measurement is lacking in Table 2, but I assume the unit used was % of CD45RA+”and “% of CD45RO+” (70% central memory cells is much too high for CD8+ T cells!) One cannot compare the amounts of naïve and effector memory cells, if different units are used in one table.
9. Please use also in Table S3 the unit “% of CD8” and “% of CD4”.
10. Please add in Fig.3 the percentage of CD25+CD127neg Treg, since these cells are the basis for “basic Tregs”, otherwise the “basic Tregs” have to be displayed as % of CD4+ T cells. Did you try to find survival differences if you use basic Tregs as % of CD4+ T cells?
11. The discussion has to be improved! The authors cite No.[24] as an example that high numbers of Treg are associated with worse prognosis (line 303). However, the paper says: “high levels of Treg cells were associated with favorable clinical outcomes”.
12. Please cite only those publications of your group which are directly related to the current investigation and not all former vaccination studies (ref. 9-15, line 63)!
13. The number of patients was complemented in Table 2, but has still to be added in Table 4.
14. Please explain SEB in the Figure legend of Fig. S2
Round 3
Reviewer 2 Report
Comments and Suggestions for Authors
The revised manuscript „ Pre-existing Immunity predicts response to first-line Immuno-therapy in Non-Small Cell Lung Cancer patients” of Xagara et al. has undoubtedly improved.
However, the authors did not change the units of naïve and memory subtypes in Table 2. Therefore, the current unit designation in Table S3 (“% in CD3CD8 T-cells” and “% in CD3CD4 T cells”) is wrong and has to be replaced. Similar as in Table 2, the units seem to be % in CD45RA+CD45RO-“ and “% in CD45RO-CD45RO+”.